# Prognostic Value of *Fusobacterium nucleatum* after Abdominoperineal Resection for Anal Squamous Cell Carcinoma

**DOI:** 10.3390/cancers14071606

**Published:** 2022-03-22

**Authors:** Marc Hilmi, Cindy Neuzillet, Jérémie H. Lefèvre, Magali Svrcek, Sophie Vacher, Leonor Benhaim, Peggy Dartigues, Emmanuelle Samalin, Julien Lazartigues, Jean-François Emile, Eugénie Rigault, Nathalie Rioux-Leclercq, Christelle de La Fouchardière, David Tougeron, Wulfran Cacheux, Pascale Mariani, Laura Courtois, Matthieu Delaye, Virginie Dangles-Marie, Astrid Lièvre, Ivan Bieche

**Affiliations:** 1Medical Oncology Department, Institut Curie, 92210 Saint-Cloud, France; hilmi.marc@gmail.com (M.H.); mdelaye@sfr.fr (M.D.); 2Digestive Surgery Department, Hôpital Saint-Antoine, AP-HP, Sorbonne Université, 75012 Paris, France; jeremie.lefevre@aphp.fr; 3Pathology Department, Hôpital Saint-Antoine, AP-HP, 75012 Paris, France; magali.svrcek@aphp.fr; 4Genetics Department, Institut Curie, 75005 Paris, France; sophie.vacher@curie.fr (S.V.); laura.courtois@curie.fr (L.C.); ivan.bieche@curie.fr (I.B.); 5Digestive Surgery Department, Gustave Roussy Institute, 94800 Villejuif, France; leonor.benhaim@gustaveroussy.fr; 6Pathology Department, Gustave Roussy Institute, 94800 Villejuif, France; peggy.dartigues@gustaveroussy.fr; 7Medical Oncology Department, Institut du Cancer de Montpellier (ICM), University of Montpellier, 34000 Montpellier, France; emmanuelle.samalin@icm.unicancer.fr; 8Gastroenterology Department, Hôpital Ambroise Paré, AP-HP, 92100 Boulogne-Billancourt, France; julien.lazartigues@curie.fr; 9Pathology Department, Hôpital Ambroise Paré, AP-HP, Université de Versailles SQY, 92100 Boulogne-Billancourt, France; jean-francois.emile@uvsq.fr; 10Gastroenterology Department, Rennes University Hospital, 35200 Rennes, France; eugenie.rigault@chu-rennes.fr (E.R.); astrid.lievre@gmail.com (A.L.); 11Medical Oncology Department, Gustave Roussy Institute, 94800 Villejuif, France; 12Pathology Department, Rennes University Hospital, 35200 Rennes, France; nathalie.rioux-leclercq@chu-rennes.fr; 13Medical Oncology Department, Centre Léon Bérard, 69000 Lyon, France; christelle.delafouchardiere@lyon.unicancer.fr; 14Department of Gastroenterology and Hepatology, Poitiers University Hospital, 86073 Poitiers, France; david.tougeron@chu-poitiers.fr; 15Medical Oncology Department, Hôpital Privé Pays de Savoie, 74000 Annemasse, France; wulfran.cacheux@curie.fr; 16Surgical Oncology Department, Institut Curie, 75005 Paris, France; pascale.mariani@curie.fr; 17Faculty of Pharmacy, Université de Paris, 75019 Paris, France; virginie.Dangles-Marie@curie.fr; 18Laboratory of Preclinical Investigation, Translational Research Department, Institut Curie, PSL Research University, 75006 Paris, France; 19Inserm U1242, COSS (Chemistry Oncogenesis Stress Signaling), Rennes 1 University, 35000 Rennes, France

**Keywords:** anal squamous cell carcinoma, intratumoral microbiota, *Fusobacterium nucleatum*, tumor biomarkers, cancer microenvironment

## Abstract

**Simple Summary:**

The main prognostic factors of localized/locally advanced anal squamous cell carcinoma (ASCC) are insufficient to predict 10–20% of metastatic relapses. *Fusobacterium nucleatum* is among the most studied bacteria in digestive tract cancers and has been described as a poor prognostic factor in several digestive cancers. We retrospectively analyzed surgical samples from a homogeneous multicenter cohort of 166 patients with ASCC who underwent abdominoperineal resection. This study showed that *F. nucleatum* was an independent predictor of favorable overall survival and disease-free survival. This allowed the identification of a patient subgroup with a good prognosis (upper tercile). Our current work strengthens the new insight into the prognostic role of intratumoral *F. nucleatum* in cancer patients. Validation of these findings would allow to guide therapeutic strategies in dedicated trials by proposing intensification or de-escalation of systemic treatments and follow-up according to *F. nucleatum* loads.

**Abstract:**

Main prognostic factors of anal squamous cell carcinoma (ASCC) are tumor size, differentiation, lymph node involvement, and male gender. However, they are insufficient to predict relapses after exclusive radiotherapy (RT) or chemoradiotherapy (CRT). *Fusobacterium nucleatum* has been associated with poor prognosis in several digestive cancers. In this study, we assessed the association between intratumoral *F. nucleatum* load and clinico-pathological features, relapse, and survival in patients with ASCC who underwent abdominoperineal resection (APR) after RT/CRT. We retrospectively analyzed surgical samples from a cohort of 166 patients with ASCC who underwent APR. *F. nucleatum* 16S rRNA gene sequences were quantified using real-time quantitative PCR. We associated *F. nucleatum* load with classical clinicopathological features, overall survival (OS), disease-free survival (DFS), and metastasis-free survival (MFS) using Cox regression univariate and multivariate analyses. Tumors harboring high loads of *F. nucleatum* (highest tercile) showed longer OS and DFS (median: not reached vs. 50.1 months, *p* = 0.01, and median: not reached vs. 18.3 months, *p* = 0.007, respectively). High *F. nucleatum* load was a predictor of longer OS (HR = 0.55, *p* = 0.04) and DFS (HR = 0.50, *p* = 0.02) in multivariate analysis. High *F. nucleatum* load is an independent favorable prognostic factor in patients with ASCC who underwent APR.

## 1. Introduction

Anal canal cancer (squamous cell carcinoma in 95% of cases) is a rare disease accounting for 2.5% of digestive cancers [1]. Human papillomavirus (HPV) infection (mainly, HPV16 and HPV18) is responsible for 90% to 95% of anal squamous cell carcinoma (ASCC) [2]. Other important risk factors include immune suppression, human immunodeficiency virus (HIV), and tobacco smoking [3].

Most patients (95%) present with local/locoregional disease at diagnosis. The aim of the treatment of localized/locally advanced ASCC is to cure the patient and achieve the best local control while maintaining a functional anal sphincter. The treatment is based on radiotherapy (RT), usually combined with 5-fluorouracil-based chemotherapy (chemoradiotherapy, CRT), and achieves approximately 80% of complete pathological response with a recurrence-free survival at 3 years of approximately 70% [4]. Surgery (abdominoperineal resection, APR) should be discussed in the cases of primary failure of RT/CRT or locoregional relapse [5]. Salvage surgery is associated with 60% of overall survival rates and 40% of relapses [6]. The treatment of metastatic relapses relies on systemic therapy (chemotherapy, immunotherapy).

The main prognostic factors of localized/locally advanced ASCC are tumor size, differentiation, lymph node involvement, HPV status, and male gender [7,8]. However, they are insufficient to predict the 10–20% of metastatic relapses that are observed after RT/CRT. Recently, it has been shown that the intestinal microbiome is associated with human diseases, including cancer [9]. Moreover, the intratumoral microbiota (i.e., bacteria found within the tumor) can also play a role in modulating carcinogenesis, immune infiltrates, and chemoresistance [10,11]. *Fusobacterium nucleatum* is among the most studied bacteria in digestive tract cancers and has been described as a poor prognostic factor in esophageal [12], gastric [13], pancreatic [14], and colorectal [15,16,17] cancers. In contrast, our team reported that high intratumoral *F. nucleatum* load was associated with longer survival in oral squamous cell carcinoma (OSCC) and was associated with a favorable immune microenvironnement [18].

In this study, we assessed the association between intratumoral *F. nucleatum* load and clinicopathological features, relapse, and survival in a homogeneous multicenter cohort of patients with ASCC who underwent APR after the failure of RT or CRT.

## 2. Materials and Methods

### 2.1. Patients

This retrospective multicenter study involved nine French centers and included all consecutive ASCC patients who underwent APR for tumor persistence or local relapse after RT or CRT from January 1996 and February 2016. We selected all patients with complete clinical and histological data and a follow-up of at least 2 years. The diagnosis of ASCC was confirmed by histology in all cases. Demographic, clinical data and tumor features, details on initial treatment by RT or CRT, indication for APR (tumor persistence or local relapse), and histological parameters from the APR were collected. After completion of RT or CRT, a persistent ulceration or a re-emergence of the anal lesion within 6 months of completion of RT was classified as persistent disease, while lesions appearing after 6 months post-RT were classified as a relapse.

Relapse was defined by the first occurrence of one of the two following events after APR: local for pelvic relapse, and metastatic for distant relapse. The study was conducted in accordance with the ethics principles of the Declaration of Helsinki and the General Data Protection Regulation (GDPR). According to French regulations, this study did not need informed consent. Patients were informed of the study by each investigator and did not express opposition.

### 2.2. Genomic DNA Extraction

For each patient, six tissue sections of 6 μm thickness were obtained from FFPE samples and a seventh tissue section was stained with HE. The tumor-rich areas were macrodissected using a single-use blade and the samples underwent proteinase K digestion in a rotating incubator at 56 °C for 3 days. DNA was extracted with the NucleoSpin kit (Macherey-Nalgen, Hoerdt, France) according to supplier recommendations. DNAs were quantified using Nanodrop spectrophotometer ND-1000 (ThermoScientific, Wilmington, DC, USA). In order to rule out external contaminations for *F. nucleatum* analysis, we included negative controls (buffers/reagents without tumor samples) and the samples were manipulated under a hood with masks and gloves.

### 2.3. Fusobacterium nucleatum Status Analysis by Real-Time Quantitative PCR

*F. nucleatum* was quantified using a real-time quantitative PCR according to the same protocol as our previous study [18]. Briefly, detection of the fluorescence signal associated with the growth of PCR products was performed and we normalized *F. nucleatum* levels on the basis of *JUN* contents [18].

HPV detection and genotyping were performed using Real-time PCR and specific primers for HPV16, and PCR to detect HPV L1 DNA and Sanger sequencing for HPV16-negative samples as previously described [19].

### 2.4. Statistical Methods

Associations among binary variables were assessed by the Chi-squared test for large samples (*n* > 60) and Fisher’s exact test for small samples (*n* < 60). Statistical significance was set at *p* < 0.05.

Loads of *F. nucleatum* are very heterogeneous among the population and one-third of the population has extremely low quantities < 0.001 (*n* = 52) (Appendix A). Cutting the population in two would have resulted in an important heterogeneity of the lowest half including patients with a difference of 100 times the loads of *F. nucleatum*. The division into four or more was not mathematically possible because two different groups would have had the same values. Therefore, *F. nucleatum* quantification was considered as terciles in order to separate the population into groups according to *F. nucleatum* loads.

Survival endpoints were defined according to the DATECAN consensus [20]. Overall survival (OS) was defined as the time from APR to death resulting from any cause. Disease-free survival (DFS) was measured from the date of APR to the time of relapse (either local or distant) or death. Metastasis-free survival (MFS) was from the date of APR to the time of metastatic relapse or death. In the absence of an event, patients were censored at the date of the last follow-up. Survival curves were estimated using the Kaplan–Meier technique and compared with the log-rank test. The Cox proportional hazard regression model was used for both univariate and multivariate analyses and for estimating the hazard ratio (HR) with a 95% confidence interval (95%CI). Prognostic factors tested in the univariate analysis were age, gender, TNM stage, type pre-operative treatment, tumor invasion depth, tumor differentiation, vascular emboli, lymphatic and perineural invasion, resection margins, and HPV status. Significative prognostic factors in the univariate analysis (*p* < 0.05) were entered into the final multivariable Cox regression model, after considering redundancy between variables. Gender and initial stage were included in the model as it is a known prognostic factor.

Univariate and multivariate Cox regression analyses and Kaplan–Meier curves were computed using the survival R package. Forest plots used for multivariate analysis were drawn through the forest model R package.

## 3. Results

### 3.1. Patient Population

From an initially established cohort of 166 collected patients with APR for persistent or recurrent ASCC after RT/CRT, 154 patients were considered for the study after the exclusion of 12 samples without information for *F. nucleatum* status. Survival analysis was restricted to 154 patients evaluable for OS and 153 patients for DFS (Figure 1).

Patient characteristics are listed in Table 1. Most patients were female (64%), aged ≤65 years old (66%), and with initial TNM tumor stages II and III (89%); 72% of them had received CRT as initial treatment.

The histological analysis of APR specimen showed a majority of lymph node-negative tumors (79%), with moderate/high differentiation (78%), vascular (61%) and lymphatic (66%) invasion, and R0 resection margins (79%). Here, 16% of tumors were associated with HIV infection, 80% with HPV16 infection and 11% were HPV-negative.

The median OS was 39.4 months from APR (64.3 months from diagnosis) and the median DFS from APR was 20.7 months.

*F. nucleatum* loads were not statistically different according to individual centers (*p* = 0.30) (Appendix A) or to the type of initial treatment (*p* = 0.49) (Appendix A).

### 3.2. Association of Fusobacterium nucleatum Load with Clinico-Pathological Features, Relapse, and Survival

High loads (upper tercile) of *F. nucleatum* were enriched in initial stage II ASCC (*p* = 0.02) and not significantly associated with other clinicopathological factors (Table 2).

One hundred and fifty-four patients were evaluable for OS. The highest tercile of *F. nucleatum* load was significantly associated with better OS compared to lower terciles (median: not reached for highest tercile vs. 50.1 months for low/intermediate terciles pooled together, *p* = 0.013) (Figure 2).

A total of 153 patients were evaluable for DFS and MFS. The highest tercile of *F. nucleatum* load was associated with better DFS compared to low/intermediate terciles (median: not reached vs. 18.3 months, *p* = 0.007) (Figure 3).

The highest and intermediate terciles of *F. nucleatum* load were associated with better MFS compared to the lowest tercile (median: 276.7 months vs. 50.1 months, *p* = 0.0054) (Appendix A). We also performed survival analyses with the diagnosis as the starting point and found once again statistically different survival rates according to *F. nucleatum* loads for OS (*p* = 0.032), DFS (*p* = 0.009), and MFS (*p* = 0.02) (Appendix A).

After excluding patients treated with RT alone, analyses performed in the subgroup of patients treated with CRT (*n* = 102) showed a significant prognostic value for OS (*p* = 0.03) and DFS (*p* = 0.015) but not MFS (*p* = 0.27).

### 3.3. Analysis of Survival Predictors

#### 3.3.1. Univariate Analysis

Positive resection margins (HR = 3.81, *p* < 0.001), lymph node invasion (HR = 3.29, *p* < 0.001), perineural invasion (HR = 2.58, *p* < 0.001), lymphatic invasion (HR = 1.72, *p* = 0.03), and tumor invasion depth (HR = 4.77, *p* = 0.01) were significantly associated with shorter OS in the univariate analysis. Age ≤ 65 years, tumor invasion depth, vascular emboli, perineural invasion, positive resection margins and lymph node invasion were also significantly associated with poor DFS (HR= 1.78, *p* = 0.02, HR = 2.71, *p* = 0.02, HR = 1.71, *p* = 0.02, HR = 1.90, *p* < 0.001, HR = 2.95, *p* < 0.001, and HR = 3.12, *p* < 0.001, respectively) in the univariate analysis. Age ≤ 65 years, perineural invasion, and lymph node invasion were associated with poor MFS (HR = 2.15, *p* = 0.02, HR = 1.90, *p* = 0.02, and HR = 4.11, *p* < 0.001, respectively) in the univariate analysis. The highest tercile of *F. nucleatum* load was significantly associated with longer OS (HR = 0.49, *p* = 0.01), DFS (HR = 0.50, *p* = 0.008) but not MFS (HR = 0.67, *p* = 0.2).

#### 3.3.2. Multivariate Analysis

In multivariate analyses, the highest tercile of *F. nucleatum* load was significantly associated with longer OS (HR = 0.55, *p* = 0.04, Figure 4A) and DFS (HR = 0.50, *p* = 0.02, Figure 4B) but not MFS (HR = 0.70, *p* = 0.29, Appendix A). However, the lowest tercile of *F. nucleatum* load was significantly associated with shorter MFS (HR = 2.25, *p* = 0.006) (Appendix A).

Among the clinicopathological parameters tested, positive resection margins and perineural invasion remained associated with poor OS, DFS, and MFS (*p* < 0.05). Age ≤ 65 years was significantly associated with shorter DFS and MFS (*p* < 0.05).

## 4. Discussion

In this work, we assessed the association between intratumoral *F. nucleatum* load and clinicopathological features and OS, DFS, and MFS in a cohort of ASCC patients who underwent APR after the failure of RT or CRT. Overall, we showed that *F. nucleatum* was an independent predictor of favorable OS, DFS, and MFS. This allowed the identification of a patient subgroup with a remarkably good prognosis (upper tercile). Other independent prognostic indicators included lymph node invasion, positive resection margins, and tumor invasion depth, as previously reported [7].

The association between *F. nucleatum* load and improved survival was unexpected as this bacteria is usually associated with poor prognosis in digestive cancers [12,13,14,15], particularly in colorectal adenocarcinoma [15,16,17]. However, these results are consistent with studies that showed that *F. nucleatum* was associated with better survival in OSCC [18,21]. OSCC and ASCC share a common histological type (i.e., squamous cell carcinoma) and are both treated with RT/CRT, while other cancers in which *F. nucleatum* showed negative prognostic effect (e.g., colorectal cancer) are adenocarcinoma and not exposed to RT. In our study, survival was analyzed by taking the date of surgery as the starting point. However, the univariate analysis goes in the same direction when we take the date of diagnosis as the starting point (Appendix A).

This positive survival effect may be mediated by modulation of intratumoral immunity [18], previously described as an independent prognostic factor in ASCC [22,23]. Data regarding the effects of *F. nucleatum* on the immune microenvironment are conflicting. Most studies showed the pro-inflammatory and immunosuppressive properties of *F. nucleatum* through expansion of myeloid-derived immune cells, Tregs, and M2 macrophages, and inhibition of cytotoxic T-cells [24,25,26,27]. On the contrary, in our previous study, we observed that OSCC tumors with high *F. nucleatum* loads were associated with a specific immune microenvironment poor in M2 macrophages, CD4 lymphocytes, fibroblasts, TLR4, OX40 ligand, and TNFRSF9, but high in TNFSF9 and IL-1ß allowing M1 polarization [18]. This suggested that intratumoral *F. nucleatum* may be associated with a tumor microenvironment insensitive to pro-inflammatory signals resulting in favorable clinical outcomes. Another recent study also reported a positive prognostic value of intratumoral *F. nucleatum* in head and neck cancers [21].

Our current work strengthens the new insight into the prognostic role of intratumoral *F. nucleatum* in cancer patients. The underlying mechanisms warrant further investigation. Yet our study has some limitations. First, we had only one cohort of patients with ASCC without a validation cohort. However, our cohort is multicentric and unique: it is the largest cohort of ASCC treated by APR and the sample size was significant given the rarity of the disease. In addition, we selected patients who required surgical intervention after the failure of RT/CRT, bringing great homogeneity to our cohort but possibly introducing a selection bias. Nevertheless, the prognostic value of a new parameter is of particular interest in a population at high risk of relapse. Finally, immune microenvironment analysis was not available in our study; it could be of interest to assess the correlation between intratumoral *F. nucleatum* expression and immune components in further studies. Besides this, *F. nucleatum* may be a predictive marker for immunotherapy response and need to be assessed in ancillary studies of immunotherapy trials. Immunotherapy is indeed being developed in ASCC but seems to be active only in a small subset of patients, and predictive biomarkers to identify patients who may benefit the most from this approach are needed.

## 5. Conclusions

In conclusion, we highlight a unique association between *F. nucleatum* and ASCC patient survival warranting further validation in larger prospective cohorts. Validation of these findings would allow to guide therapeutic strategies in dedicated trials by proposing intensification or de-escalation of systemic treatments and follow-up according to *F. nucleatum* loads. This can also give a rationale for further exploration of the role *of F. nucleatum* in ASCC carcinogenesis and response to treatment, particularly immunotherapy.

## Figures and Tables

**Figure 1 cancers-14-01606-f001:**
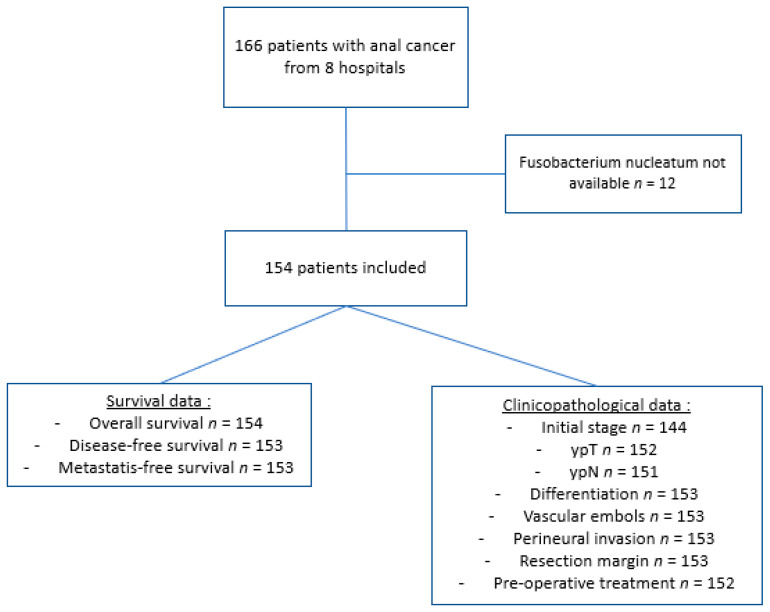
CONSORT-like workflow diagrams for the cohort.

**Figure 2 cancers-14-01606-f002:**
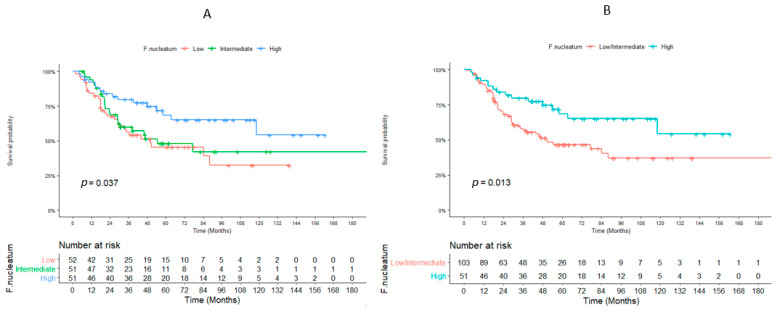
Association between overall survival and *Fusobacterium nucleatum.* Overall survival curves for the *Fusobacterium nucleatum* divided in 3 categories according to terciles (**A**) and 2 categories according to terciles (**B**), *n* = 154 patients.

**Figure 3 cancers-14-01606-f003:**
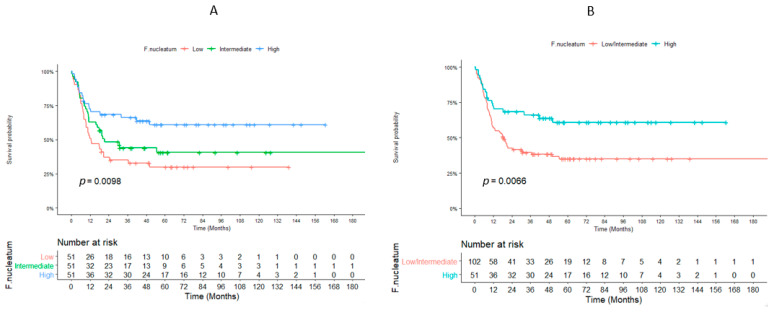
Association between disease-free survival and *Fusobacterium nucleatum*. Disease-free survival curves for the *Fusobacterium nucleatum* divided into three categories according to terciles (**A**) and two categories according to terciles (**B**), *n* = 153 patients.

**Figure 4 cancers-14-01606-f004:**
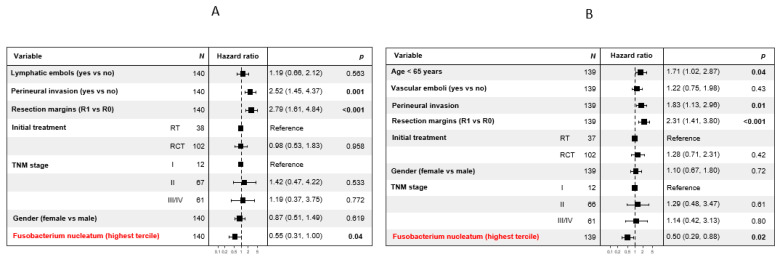
Prognostic value of clinicopathological factors and *Fusobacterium nucleatum*. Multivariate analysis for the clinicopathological factors regarding overall survival ((**A**), *n* = 148 patients) and disease-free-survival ((**B**), *n* = 147 patients). Bolded *p*-values are significant (<0.05).

**Table 1 cancers-14-01606-t001:** Initial patient characteristics and histological parameters from abdominoperineal resection.

Variables	*N*	%
Total	154	100
Age at the time of diagnosis, years		
≤65	101	65.6
>65	53	34.4
Gender		
Female	98	63.6
Male	56	36.4
TNM stage at the time of diagnosis		
Stage I	14	9.7
Stage II	68	47.2
Stage III	60	41.7
Stage IV	2	1.4
Pre-operative treatment		
Radiotherapy	43	28.3
Chemoradiotherapy	109	71.7
ypT (tumor invasion depth)		
ypT1	20	13.2
ypT2	54	35.5
ypT3	34	22.1
ypT4	44	28.9
ypN		
ypN0	120	78.9
ypN+	32	21.1
Tumor differentiation *		
Low	33	21.6
Moderate/high	120	78.4
Vascular emboli		
Yes	93	60.8
No	60	39.2
Lymphatic invasion		
Yes	100	65.4
No	53	34.6
Perineural invasion		
Yes	88	57.5
No	65	42.5
Resection margin		
R0	121	79.1
R1	32	20.9
HPV status		
Negative	17	11
Serotype 16	123	79.9
Serotype 18	4	2.6
Other serotypes	10	6.5
HIV status		
Negative	117	84.2
Positive	22	15.8

* according to the AJCC Cancer Staging Manual.

**Table 2 cancers-14-01606-t002:** Association between Fusobacterium nucleatum and clinicopathological factors.

Variables	*F. nucleatum* Low/Intermediate		*F. nucleatum* High		*P*
	N	%	N	%	
Total	103	66.9	51	33.1	
Age, years					
≤65	65	63.1	36	70.6	0.46
>65	38	36.9	15	29.4	
Gender					
Female	68	66	30	58.8	0.49
Male	35	34	21	41.2	
Initial stage					
Stage I	11	11.1	3	6.7	0.02
Stage II	41	41.4	27	60	
Stage III	47	47.5	13	28.9	
Stage IV	0	0	2	4.4	
ypT					
ypT1	15	14.9	5	9.8	0.3
ypT2	37	36.6	17	33.3	
ypT3	18	17.5	16	31.4	
ypT4	31	30.7	13	25.5	
ypN					
ypN0	78	76.5	42	84	0.4
ypN+	24	23.5	8	16	
Differentiation					
Low	19	18.6	14	27.5	0.3
Moderate/high	83	81.4	37	72.5	
Vascular emboli					
Yes	60	58.8	33	64.7	0.6
No	42	41.2	18	35.3	
Lymphatic invasion					
Yes	70	68.6	30	58.8	0.31
No	32	31.4	21	41.2	
Perineural invasion					
Yes	58	56.9	30	58.8	0.95
No	44	43.1	21	41.2	
Resection margin					
R0	79	77.5	42	82.4	0.62
R1	23	22.5	9	17.6	
HPV status					
Negative	10	9.7	7	13.7	0.63
Serotype 16	84	81.6	39	76.5	
Serotype 18	2	1.9	2	3.9	
Other serotypes	7	6.8	3	5.9	

## Data Availability

The datasets used and/or analyzed during the current study are available from the corresponding author on reasonable request.

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
