# Peer review of "Prognostic Value of Fusobacterium nucleatum after Abdominoperineal Resection for Anal Squamous Cell Carcinoma"

_cancers, 2022, doi:10.3390/cancers14071606_

Round 1

Reviewer 1 Report

My comments on the original paper were fairly minor compared to issues raised by the editor and other reviewers. In regard to the other reviewers, the issues raised seem to have been addressed as much as can be done without delaying publication awaiting a new set of patients with possible revisions in the study protocol. The great effort in putting this study together seems to balance any flaws that may exists in the current paper. I do note that my apparent confusions over met-free survival was based on mislabeling of old SFig 2 and now SFig 5,which is still mislabeled, showing " highest tercile" for met-free survival instead of "lowest tercile". Also, in terms of presentation, I don't think that SFigs1 and 2 are adequate as presentation of the distribution of F, nucleatum levels. Maybe a Log plot would be better, so we could see the low end distributions and the plot could show the splits among the terciles. And, I don't see why all of the hazard ratio analysis doesn't emphasize the increased risk for the lowest tercile set (presentation as in "corrected labeling on terciles" S Fig 5) as the problem is initially defined as predicting risk for "the 10-20% of metastatic relapses " (lines 81-84). There is also a question raised by Table 2 over F, nucleatum loads vs. Stage. Does bacterial load really show up better in term of prognosis than stage?

Author Response

Paris, 11th March 2022

Dear Reviewer,

We would like to thank you for your very helpful comments and suggestions.

We have performed the revision of our manuscript according to the latest comments.

Comments have been highlighted in bold type; answers and modifications to the manuscript were written in red font.

My comments on the original paper were fairly minor compared to issues raised by the editor and other reviewers. In regard to the other reviewers, the issues raised seem to have been addressed as much as can be done without delaying publication awaiting a new set of patients with possible revisions in the study protocol. The great effort in putting this study together seems to balance any flaws that may exists in the current paper.

We thank the reviewer for the positive appreciation.

Also, in terms of presentation, I don't think that SFigs1 and 2 are adequate as presentation of the distribution of F, nucleatum levels. Maybe a Log plot would be better, so we could see the low end distributions and the plot could show the splits among the terciles.

We changed the Supplementary Figure S1 and S2 with logged values in order to show the low-end distribution. As mentioned in the Materials and Methods section, cutting the population in two would have resulted in an important heterogeneity of the lowest half and the division in 4 or more was not mathematically possible because 2 different groups would have had the same values. Therefore, F. nucleatum quantification was considered as terciles in order to separate the population into groups according to F. nucleatum loads.

I do note that my apparent confusions over met-free survival was based on mislabeling of old SFig 2 and now SFig 5,which is still mislabeled, showing " highest tercile" for met-free survival instead of "lowest tercile". And, I don't see why all of the hazard ratio analysis doesn't emphasize the increased risk for the lowest tercile set (presentation as in "corrected labeling on terciles" S Fig 5) as the problem is initially defined as predicting risk for "the 10-20% of metastatic relapses " (lines 81-84).

Thank you for this remark. We compared the highest tercile vs other terciles due to the very close prognosis of the low and intermediate terciles as visualized in the Figure 2 and 3. Identifying a good prognosis group is also part of predicting the risk of relapse/death.

We are sorry for the mistake in the Figure S5. Only for MFS, comparing the highest tercile vs other terciles was not significant (p=0.29). In order to keep consistency with other forest plots (Fig 4), we added a new panel in the Figure S5 showing the highest tercile as reference. We also modified the manuscript to clarify the interpretation of MFS:

Line 224: “In multivariate analyses, the highest tercile of F. nucleatum load was significantly as-sociated with longer OS (HR=0.47, p=0.016, Figure 4A) and DFS (HR=0.48, p=0.01, Figure 4B), but not MFS (HR=0.70, p=0.29, Figure S5A). However, the lowest tercile of F. nucleatum load was significantly associated with shorter MFS (HR=1.82, p=0.04) (Figure S5B).”

There is also a question raised by Table 2 over F, nucleatum loads vs. Stage. Does bacterial load really show up better in term of prognosis than stage?

We agree with this remark and included the TNM staging into the multivariate analyses. Our results remain significant after adjusting with this factor (new Figure 4 and Figure S5).

We hope that these revisions improve the manuscript such that you deem it suitable for publication.

Once again, we thank you for the interest in our work.

Best regards,

Marc Hilmi

Cindy Neuzillet

Astrid Lièvre

Reviewer 2 Report

After the correction I think the paper merits publication.

Author Response

We thank the reviewer for this positive comment.

Round 2

Reviewer 1 Report

Set to publish

Author Response

We thank the Reviewer for improving our manuscript.

This manuscript is a resubmission of an earlier submission. The following is a list of the peer review reports and author responses from that submission.

Round 1

Reviewer 1 Report

This paper is very excellent and can be acceptable.

Reviewer 2 Report

In an attempt to identify better predictors for metastatic relapses of anal squamous cell carcinoma (ASCC), the authors report in this manuscript (MS ID 1472173) results of a retrospective analysis of surgical samples collected after abdominoperineal resection (APR) from 166 ASCC patients. The study focused on determining the prevalence of Fusobacterium nucleatum to evaluate its possible prognostic value. The results included in this manuscript show that the tumor samples with the top-third highest F. nucleatum load corresponded to patients who had longer Overall Survival (OS) and Disease-free Survival (DFS), and these findings are taken by the authors to conclude that high levels of F. nucleatum is a favorable prognostic factor for ASCC patients who underwent APR.

Research to identify molecular or biological markers that may have predictive value towards determining the course of the progression or relapse of any type of tumor is highly relevant, and it may lead to useful, actionable treatment strategies. From that point of view, the research reported in this manuscript appears to be meritorious. However, there is a key point of concern: the fact that the finding of high F. nucleatum load as a favorable prognosis factor goes against the vast majority of findings reported in the literature with regard to a wide variety of other tumor types.

The authors state that their studied patient population is unique on the basis of: (1) the size of the patient sample, and (2) their own appraisal as a homogeneous multicenter cohort. However, it is important that this “homogeneous” nature of the cohort be demonstrated, particularly when the samples derive from multiple centers, which could be an intrinsic source of heterogeneity. Differences related to the procedures followed for sample collection, handling and processing among the various institutions may lead to artefactual disparities, which may ultimately confound the results and the final data interpretation. Therefore, the manuscript should include information on the distribution of F. nucleatum load in samples from each individual center participating in the study, to allow the detection of any possible center-derived bias (e.g., all the high-load samples coming from one place).

In addition, there are several minor changes that need to be introduced in the manuscript:

(1) Fusobacterium nucleatum, similar to the scientific name of any other species must be in italics throughout the manuscript.

(2) F. nucleatum is not an abbreviation of the full bacterium name. Therefore, it is incorrect to write “Fusobacterium nucleatum (F. nucleatum)”, as it appears in several places within the manuscript. The full bacterium name should be use only the first time it is mentioned in the text, and F. nucleatum every time afterwards, and always using a font in italics.

(3) The number of participating centers needs to be clarified, as the Materials and Methods section 2.1 talks about “9 French centers”, while Figure 1 refers to “8 hospitals”.

(4) The criteria used to determine the degree of differentiation of the various tumors must be clearly stated.         

Reviewer 3 Report

I thank the authors for the opportunity to review their manuscript, entitled "Prognostic value of Fusobacterium nucleatum after abdom-2 inoperineal resection for anal squamous cell carcinoma".

Overall, the authors present an interesting research question regarding the impact of F.Nucleatum tissue load on prognosis post APR for relapsed/resistant disease.

I have included several comments and suggestions:

  1. Introduction:
    1. Please include data on ASCC survival, recurrence rates
  2. Materials and Methods
    1. Can OS/DFS/MFS be calculated from timepoint of diagnosis? if not, please include data of timespan between diagnosis to APR. Is it possible that timespan from Dx to APR is a prognostic marker that should have been included?
    2. How were terciles chosen at cutoffs? is it based on a published metric, ROC or were other thresholds also assessed and discarded?
    3. It is important to note that the authors calculated DFS and MFS as the time to event OR Death (which means that a pure time to event might result in other data and other conclusions).
  3. Results
    1. 2 patients with Stage 4 disease are included in the analysis - would consider excluding as DFS and MFS should not be calculated in these cases anyway.
    2. Were other parameters also assessed in the univariate analysis - male gender, type of previous therapy, indication for APR (Persistent vs. Relapsed), HPV status. As Male gender is a known factor, it would be interesting to see it included in the models as well. 
  4. Discussion
    1. The authors mention positive correlation between load and survival - however the statistical analysis employed only tests for an association and not correlation.
    2. It is interesting to see if there are data regarding F.Nucleatum at time of Dx (maybe from the biopsies?). Is it possible that colonization varies during the treatment period? 
    3. It  important to note that the methodology of the study does not include the time from initial diagnosis but  just from the point of APR.    

Reviewer 4 Report

This study attempts to validate F. nucleatum detection in anal squamous cell carcinoma as prognostic marker for potential relapse after initial treatments.  The examination of bacteria as a contributing factor in cancer is interesting.  However, I fail to see how F. nucleatum levels help in identifying a vulnerable subpopulation any more than the existing prognostic tools.  Then there is the odd finding that metastasis-free-survival is associated with the opposite tercile vs. overall survival and disease free survival.  I need a more compelling analysis and discussion to be convinced that this tool would really help the clinicians.